# Naphthoquinone preferentially pairs with non-proton-pumping NADH dehydrogenase for respiratory electron transport

Snehal V. Khairnar, Anjali V. Patil, L. Karvannan, Amitesh Anand ⓘ *

Department of Biological Sciences, Tata Institute of Fundamental Research, Mumbai, Maharashtra, India

* amitesh.anand@tifr.res.in

**Editor:** Danielle A. Garsin, University of Texas McGovern Medical School: The University of Texas Health Science Center at Houston John P and Katherine G McGovern Medical School, UNITED STATES OF AMERICA

## Abstract

Optimal resource allocation is crucial to bacterial physiology and necessitates strategic metabolic decisions. One such evolutionary adaptation was the shift to high-potential respiratory chains following Earth's Great Oxidation Event. Respiratory quinones, key redox-active electron carrier molecules, evolved from naphthoquinones (NQs) to ubiquinones (UQs) in response to oxygen availability. The two quinone types differ in their redox potential, with UQs possessing higher potential. Therefore, NQs are more autooxidizable and electron-leaky than UQs. Using adaptive laboratory evolution of a NQ-dependent *Escherichia coli* strain, we previously showed the fitness advantage of high-potential quinones. Here, we resolve a paradoxical growth benefit conferred by the loss of function of the pyruvate dehydrogenase complex regulator, revealing that NQs preferentially pair with the non-proton-pumping NADH dehydrogenase, thereby optimizing electron transport in low-potential respiratory chains under aerobic conditions.

## Author summary

Energy generation is central to life, and organisms adjust their metabolism to match their environment and maintain efficiency. This metabolic balancing act is especially prominent in bacteria. During Earth's Great Oxidation Event, rising oxygen levels forced microbes to rewire their energy metabolism. A major change was in the molecules that shuttle electrons in respiration—respiratory quinones. Many bacteria evolved the ability to synthesize ubiquinone, a high reduction potential quinone that performs better in oxygen-rich environments. We investigated the genetic strategies behind the metabolic adaptation required to respire using the more ancient quinone, naphthoquinone, under aerobic conditions. In naphthoquinone-dependent *Escherichia coli*, we observed precise genetic changes that exert opposite transcriptional effects on two enzymes within the same regulatory network, enabling growth without disrupting resource

**Data availability statement:** DNAseq and RNAseq data supporting this study are deposited in the NCBI SRA (PRJNA1294558) and GEO (GSE303520).

**Funding:** This work was supported by the Science and Engineering Research Board, Department of Science and Technology, India (SRG/2023/001921 to AA). The funders had no role in study design, data collection and analysis, decision to publish, or preparation of the manuscript.

**Competing interests:** The authors have declared that no competing interests exist.

balance. We further establish a functional synergy between naphthoquinone and the non-proton-pumping NADH dehydrogenase. These findings reveal a crucial design principle in microbial energy metabolism that shaped bacterial evolution.

## Introduction

Cellular resource partitioning is critical to physiological adaptation, and achieving such optimality requires careful metabolic choices [1–4]. While the dynamic lifestyle of bacteria continuously requires such adjustments, a few metabolic adaptations occurred on an evolutionary scale. A significant adaptive evolution in energy metabolism has been the shift towards a high-potential respiratory chain in response to the Great Oxidation Event (GOE) on Earth [5,6].

Respiratory quinones are elegantly designed redox-active molecules where the hydrophobic isoprenoid chain enables diffusion within the membrane, and the redox-active ring with two oxo groups facilitates the electron shuttling between the enzymes of the electron transport system (ETS) (Fig 1A and 1B). The two prevalent quinone types differ in their redox-active head group: naphthalene-1,4-dione in naphthoquinones (NQs: menaquinone and demethylmenaquinone) and 1,4-benzoquinone in ubiquinones (UQs). UQ is the canonical respiratory quinone in aerobic ETS. These two functionally similar quinone types date to pre- and post-GOE, respectively.

We recently demonstrated the metabolic significance of high-reduction-potential quinones, such as UQ, over NQ in oxygen-rich environments by performing adaptive laboratory evolution of a NQ-dependent *Escherichia coli* strain under aerobic conditions [7]. However, the physiological relevance of the loss of function mutation fixed in the transcriptional repressor pyruvate dehydrogenase complex regulator (PdhR) remained elusive [8]. PdhR represses the non-proton-pumping type-II NADH dehydrogenase (NDH-2). However, lack of PdhR relieves repression of another regulatory target- pyruvate dehydrogenase complex (PDC), leading to its upregulation and constraining the allocation of resources toward a growth-supportive proteome [9]. Therefore, the growth improvement of the NQ-dependent strain imparted by the loss of function of PdhR creates a mechanistic paradox. Here, we resolve this metabolic conflict and establish the preferred pairing of NQ with the NDH-2.

## Results & discussion

We generated an *E. coli* strain (Δ*menF*Δ*ubiC*) that should be deficient in the biosynthesis of respiratory quinones by genetically deleting chorismate lyase (*ubiC*) and isochorismate synthase (*menF*). UbiC and MenF are involved in the biosynthesis of UQ and NQ, respectively (Fig 1B). However, we observed that the growth of the strain is equivalent to that of the NQ-dependent strain (Δ*ubiC*), potentially due to enzymatic redundancy between the two isochorismate synthases- MenF and EntC [7, 10] (Fig 1C). These observations consolidate the redundancy of the isochorismate synthases in *E. coli* metabolism [11].

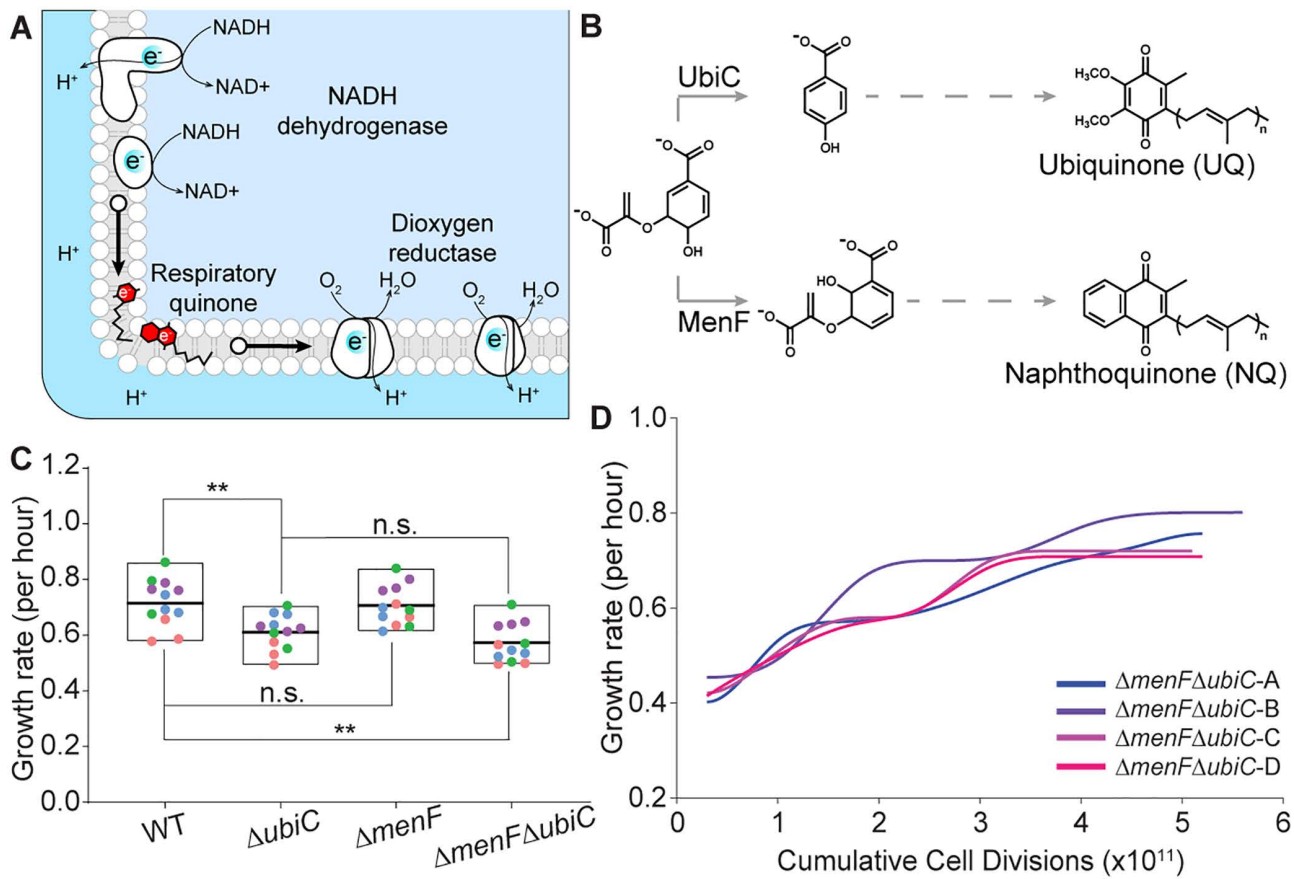

**Fig 1. Growth behavior of naphthoquinone-dependent *E. coli*. (A)** A simplistic scheme for the *E. coli* aerobic ETS working between NADH and oxygen. Electrons donated by NADH travel to dioxygen reductase mediated by respiratory quinones and finally accepted by oxygen. **(B)** Chorismate lyase (UbiC) and isochorismate synthase (MenF) acting at chorismate node for the biosynthesis of ubiquinone and naphthoquinone, respectively. **(C)** Growth rates of strains with presence or absence of the isochorismate synthase and/or chorismate lyase under aerobic conditions. Horizontal lines indicate the mean from four biological replicates (each with three technical replicates, color-coded), and vertical bars show the range (minimum to maximum). Statistical significance was assessed using the Kruskal–Wallis test. ** $p < 0.01$, n.s. (not significant) $\geq 0.05$. **(D)** Evolutionary trajectories for the growth improvement of the $\Delta menF\Delta ubiC$ strain. The evolution was performed with four independent lineages.

We proceeded with performing adaptive laboratory evolution of $\Delta menF\Delta ubiC$ to examine the evolutionary outcomes. We evolved four independent lineages of $\Delta menF\Delta ubiC$ under aerobic conditions and observed an improvement in their growth rate in about $5 \times 10^{11}$ cumulative number of cell divisions (Fig 1D). Hereafter, prefix 'u' and 'e' before strain genotypes designate "unevolved" and "evolved," respectively.

We performed whole-genome sequencing of evolved strains to examine growth-promoting genetic changes. Three of the four independently evolved lineages fixed mutations specifically in the PdhR box upstream of the *ndh* gene (encoding NDH-2) (Fig 2A). We observed an increased expression of *ndh*, indicating that the mutation alleviated PdhR repression on *ndh* (Fig 2A and 2B). This observation enabled us to identify *ndh* as the specific PdhR regulon member under adaptive selection pressure for NQ-dependent strains.

The fourth lineage showed a mutation in the DNA-binding domain of PdhR, consistent with previous observations [7]. The transcriptional outcomes of the mutations in the PdhR box of *ndh* suggest that the PdhR open reading frame mutation would be a loss of function mutation. However, a systemic reduction in the functioning of PdhR will cause upregulation of

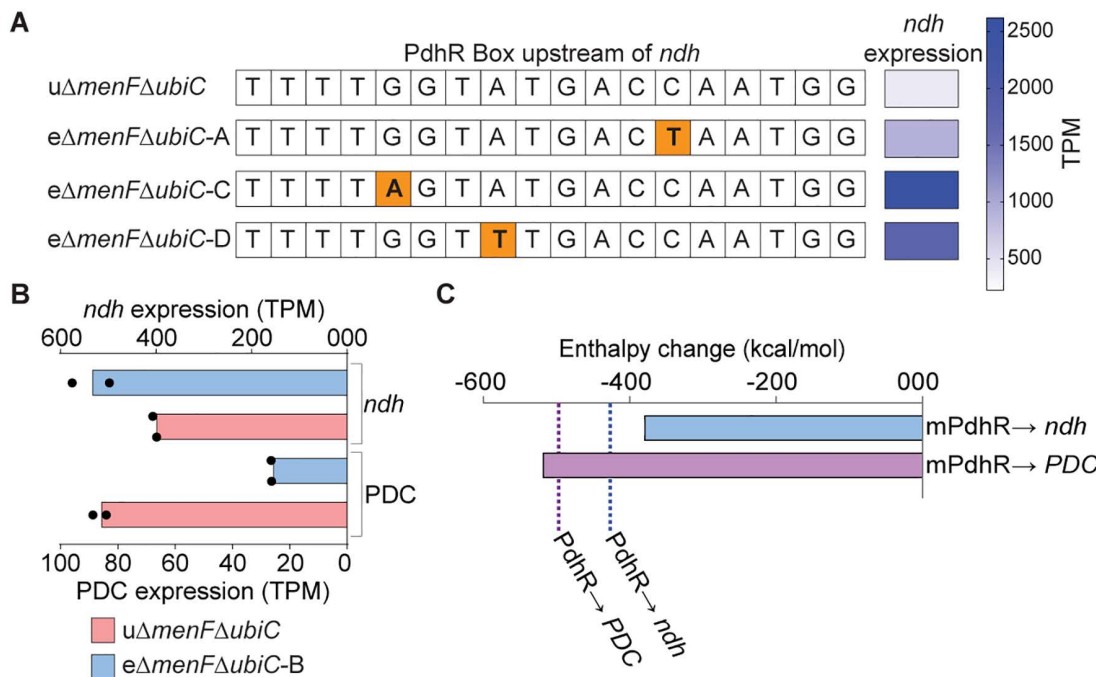

**Fig 2. Growth optimization strategies of naphthoquinone-dependent *E. coli*. (A)** The common mutations observed in three evolved lineages of Δ*menF*Δ*ubiC* and their impact on the expression of the gene encoding NDH-2 (*ndh*). The mutation location is highlighted in orange. **(B)** Expression of PDC (based on expression levels of the three subunits of this complex) and *ndh* in the evolved Δ*menF*Δ*ubiC*-B lineage. **(C)** Estimated enthalpy changes from protein–DNA docking between mutant PdhR (mPdhR) or WT PdhR and the *pdhR* box regions of *ndh* and PDC. Dashed lines indicate enthalpy values for WT PdhR–DNA complexes.

PDC, resulting in growth retardation [9]. To resolve this cost-benefit paradox, we performed protein–DNA docking simulations. The PdhR box of PDC and *ndh* differs in nucleotide composition in their non-consensus region [8]. We therefore examined the binding of WT and mutated PdhR to the two PdhR boxes. As compared to WT PdhR, the mutated PdhR appears to bind to the PdhR box of PDC more efficiently; however, its binding to the *ndh* PdhR box is compromised (Fig 2C). This differential perturbation in binding properties explains the downregulation of PDC while upregulating *ndh* (Fig 2B). Encouragingly, the strain design inadequacy resolved the conflict between proteome imbalance resulting from PdhR loss of function and its adaptive fixation in the NQ-dependent strain during growth rate improvement.

Additionally, all evolved lineages fixed well-characterized mutations in RNA polymerase subunit or in the intergenic region between *pyrE* and *rph* (S1 Table). These mutations provide a growth advantage on the media used during this adaptive laboratory evolution [12]. While there was no other common mutation observed in the evolved strains, two notable mutations were in pyruvate kinase (*pykF*) and methyltransferase (*ubiE*). Given these mutations were present in more than one evolved lineage, we performed growth comparison between strains with and without these mutations. While we can not completely rule out any additional benefit provided by these mutations, we did not observe any growth difference between strains with and without these genetic changes (S1 Fig).

The evolved Δ*menF*Δ*ubiC* strain showed an increase in oxygen consumption rate, suggesting an improvement in aerobic respiratory capacity (Fig 3A). The upper ETS is recognized as the primary source of cellular reactive oxygen species; therefore, optimal functioning of this part is critical for energy and redox homeostasis [13]. We observed a maintained $NAD^+$/NADH ratio in unevolved and evolved strains (Fig 3B). This redox maintenance with electron-leaky respiratory quinone may be the growth bottleneck in the unevolved strain, and upregulation of the high-turnover dehydrogenase

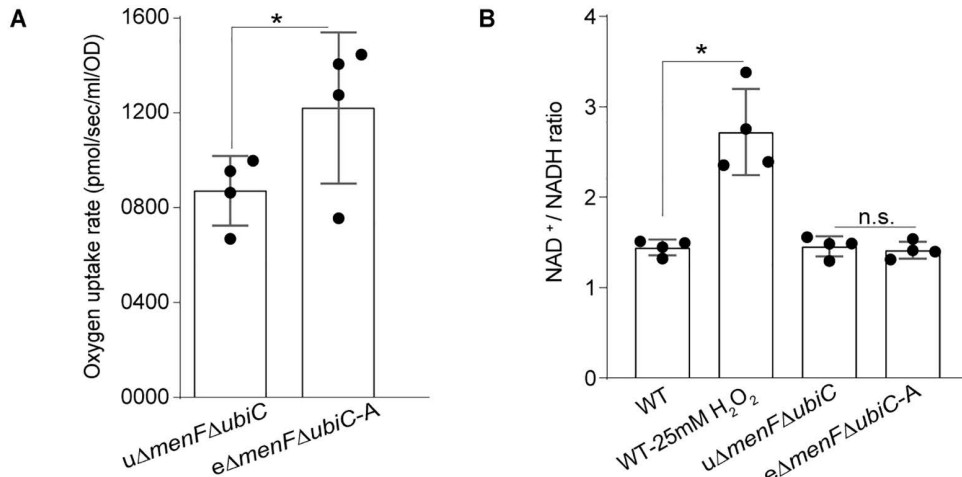

**Fig 3. Metabolic changes post adaptive laboratory evolution of ΔmenFΔubiC.** **(A)** Oxygen uptake rates of uΔmenFΔubiC and eΔmenFΔubiC-A strains, values for all four biological replicates are shown. Error bars represent standard deviation. Statistical significance was assessed using the Mann–Whitney U test. **(B)** NAD+/NADH ratio in wild-type and NQ-dependent strains. WT cells were treated with $H_2O_2$ as positive controls for the assay. Error bars represent standard deviation. Statistical significance was assessed using the Kruskal–Wallis test. * $p < 0.05$, n.s. (not significant) $\geq 0.05$.

NDH-2 may alleviate this redox stress [14]. The transfer of electrons from proton-pumping type-I NADH dehydrogenase (NDH-1) to respiratory quinones takes place in two steps involving the formation of a semiquinone free radical. These radicals are prone to generating reactive radicals by direct interaction with oxygen. However, unlike the chain of iron-sulfur clusters in NDH-1, NDH-2 uses a highly co-operative flavin co-factor that facilitates a two-electron transfer from NADH to quinone [15, 16]. Therefore, the higher catalytic efficiency of NDH-2 for NQ and reduced semiquinone generation can provide a safer route for the electron flow [17]. Furthermore, the lower reduction potential of NQ as compared to UQ means lower driving Gibbs free energy for electron transfer. This thermodynamic difference makes NDH-2 a better metabolic partner for NQ, as proton-uncoupled electron transfer by this dehydrogenase reduces the chances of reverse electron flow [18,19].

The involvement of NDH-2 in the growth improvement of NQ-dependent strain and their potential functional synergy motivated us to explore the concerted distribution of respiratory quinones (UQ and NQ) and NADH dehydrogenases (NDH-1 and NDH-2). We developed custom hidden Markov Model profiles for the genes suggestive of the presence of corresponding respiratory quinone biosynthesis and NADH dehydrogenase (S2 Table) [20]. We generated these position-specific probabilistic models using sequence clustering approach developed earlier to reduce any bias in our search due to over-representation of commonly studied and sequenced bacterial species [21]. We used these profiles to search for the co-conservation of the biosynthetic pathways of respiratory quinones with the NADH dehydrogenases across approximately 9000 bacterial species (Fig 4A, S1 Sheet). The majority of bacterial species (65.4%) with the NQ biosynthetic pathway showed the presence of NDH-2, either exclusively or in combination with NDH-1. Only a small fraction of bacterial species (0.9%) exhibited the coexistence of NDH-2 with UQ.

The overwhelming co-occurence of NDH-2 with NQ biosynthetic pathway in bacterial species encouraged us to analyze the impact of the absence of NDH-1 and NDH-2 on the growth profile of the NQ-dependent strain. While the deletion of subunit B of NDH-1 (nuoB) did not impact the growth of ΔmenFΔubiC, the deletion of the gene encoding NDH-2 (ndh) caused a significant drop in growth (Fig 4B). Together, these observations support metabolic compatibility between naphthoquinone and NDH-2.

PLOS Genetics

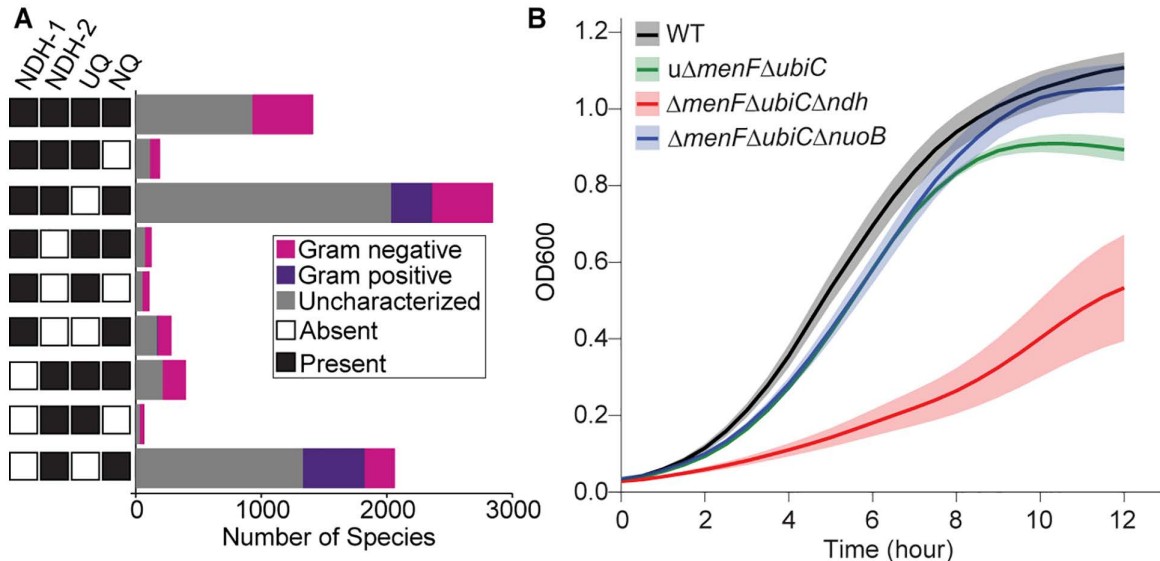

**Fig 4. Synergy between type-II NADH dehydrogenase and naphthoquinones. (A)** Bar plot showing the distribution of NADH dehydrogenases and respiratory quinones across 7783 bacterial species. UQ is ubiquinone and NQ is naphthoquinone. **(B)** Aerobic growth profile of NQ-dependent *E. coli* strains lacking either NDH-2 (Δ*menF*Δ*ubiC*Δ*ndh*) or NDH-1 (Δ*menF*Δ*ubiC*Δ*nuoB*). Values represent means from four biological replicates (each with three technical replicates); shaded bands indicate standard error of the mean.

In conclusion, we report our observation of targeting the regulatory element enabling specific expression increase in NDH-2 in strains dependent on NQ for respiration. We also demonstrate the evolutionary co-conservation of these two components of ETS. The identification and characterization of metabolic partners of type-II NADH dehydrogenase further facilitate the effective targeting of this enzyme for antimicrobial development.

## Materials and methods

### Bacterial strains and growth conditions

Experiments were conducted using *Escherichia coli* K-12 MG1655 (ATCC 700926) as the wild-type strain. Gene deletions were performed by homologous recombination with recombination cassettes derived from the Keio collection. P1 phage was used for the transduction and recombination [22]. Gene knockouts were confirmed by colony PCR screening and whole-genome resequencing. Media components were purchased from Sigma-Aldrich. All the growth assays were performed using the Tecan Spark multimode microplate reader at 37$^{\circ}$C and atmospheric oxygen partial pressure using 96-well plates with 200µL culture per well. Four biological replicates (each with three technical replicates) were performed for each experiment. GCplyr (R package for microbial growth analysis) was used to determine the growth rate [23].

### Adaptive laboratory evolution, DNA and RNA resequencing

Adaptive laboratory evolution (ALE) of Δ*menF*Δ*ubiC* strain was carried out with four independent lineages. Cultures were cultivated in M9 minimal medium supplemented with 4 g/L glucose at 37°C under continuous agitation, to maintain optimal aeration and avoid nutrient depletion. Cultures were periodically passaged to fresh flasks once the optical density at 600 nm (OD$_{600}$) reached mid to late log phase. The evolution proceeded under conditions designed to maintain cells in exponential growth for extended durations. Once cells reached their maximum growth rate, evolution was stopped. A growth rate trajectory was constructed from serial OD measurements, using the natural logarithm of OD$_{600}$ over time. A smooth, monotonic cubic spline curve was fit to growth data to estimate fitness gains.

End-point isolates, which are termed evolved strains, from each evolutionary lineage were subjected to genome rese-quencing for mutation analysis as described previously [10]. For expression analysis, RNA was extracted from strains grown under conditions similar to those used during ALE. Two biological replicates were analyzed per strain. RNAseq was performed as described previously [10].

## Oxygen uptake characterisation

The oxygen uptake of each culture was determined by measuring the depletion of dissolved oxygen using the Oroboros O2K system. The cultures were grown in M9 minimal medium supplemented with 4 g/L glucose at 37°C and 300 rpm, and oxygen uptake rates were normalized to $OD_{600}$.

## $NAD^+$/NADH Ratio estimation

The intracellular $NAD^+$/NADH ratio was measured using the Promega NAD/NADH-Glo Assay Kit (G9071) following the man-ufacturer's protocol. Briefly, *E. coli* strains were grown to mid-log phase, and cells equivalent to OD600~1 were harvested for $NAD^+$ and NADH extraction and quantification. *E. coli* WT treated with 25mM $H_2O_2$ was used as a positive control for the assay.

## Co-conservation analysis of NADH dehydrogenases and respiratory quinones biosynthesis genes

To examine the co-conservation of NADH dehydrogenases and respiratory quinone biosynthesis pathways, we ana-lyzed 9,433 bacterial genomes. Representative genes indicative of specific dehydrogenases or respiratory quinone biosynthetic pathways were [24]: (i) classical (*menB, menC, menF*) NQ biosynthesis, (ii) futalosine (*mqnA, mqnC, mqnD*) NQ biosynthesis pathway, (iii) UQ biosynthesis pathways (*ubiA, ubiC, ubiG*), (iv) NDH-1 subunits (*nuoA, nuoE, nuoJ*), and (v) NDH-2 (*ndh*). The gene search was performed by building HMM profiles as described previously, with sequence clustered at 50% identity to remove redundancy [25]. Hits were filtered using gene-specific e-value thresh-olds and a minimum Bit score of 100. The filtered data were used for downstream co-conservation analysis.

## Protein–DNA docking of mutant PdhR

Protein–DNA docking simulations were performed using HADDOCK 2.4 [26]. The structure of wild-type PdhR was obtained from the Protein Data Bank (PDB ID: 5KVR), and the PdhR-box DNA sequences from the promoter regions of PDC and *ndh* were retrieved from RegulonDB [27]. Three-dimensional models of wild-type and A2S mutant PdhR–DNA complexes were generated using AlphaFold3 [27, 28]. Each modeled complex was split into separate protein and DNA PDB files. Standard preprocessing, which involves assignment of chain IDs and structural verification, was performed using PyMOL [29]. Ambiguous interaction restraints (AIRs) were defined based on conserved DNA-binding motifs. Dock-ing proceeded in three stages—rigid-body (it0), semi-flexible (it1), and water refinement—with 1000, 200, and 200 models generated, respectively, using default HADDOCK parameters. Top-scoring clusters were selected based on HADDOCK score, and binding affinities were compared using the ΔH energy metric:

$$\Delta H = E_{complex} - (E_{protein} + E_{DNA})$$

## Statistical analysis

Prism (GraphPad) version 10 was used for design and analysis of the plots. All experiments in this study were performed at least in triplicate. The growth rate plots show each of the four biological replicates (each with three technical replicates), and the error bars show the standard deviation in the obtained data. The non-parametric Kruskal-Wallis test was per-formed to determine the significance of growth rate differences. The significance for the oxygen uptake rates was deter-mined using the Mann–Whitney U test. The figure legends of each plot mention the statistical details and tests used.

## Supporting information

**S1 Fig. Growth profile of ΔmenFΔubiC strains to assess influence of additional mutations observed during adaptive evolution.** (A) The lineage B has a point mutation in *ubiE* but grows similar to lineage A that has wild type *ubiE* sequence. (B) The lineage B has a point mutation in *pykF* but grows similar to lineage D that has wild type *pykF* sequence. Values represent means from four biological replicates (each with three technical replicates); shaded bands indicate standard error of the mean.
(TIF)

**S1 Table. List of mutated genes in strains isolated at the endpoint of the evolutionary trajectory.**
(PDF)

**S2 Table. The table summarizes the representative genes associated with specific respiratory quinone biosynthesis pathways and NADH dehydrogenases and their custom e-value cutoffs.**
(PDF)

**S1 Sheet. List of bacterial genomes used in this study with search results of representative genes involved in naphthoquinone and ubiquinone biosynthesis pathways, and NADH dehydrogenases.**
(XLSX)

**S2 Sheet. Numerical data for graphs and summary statistics.**
(XLSX)

## Acknowledgments

We thank Prof. Bernhard O. Palsson, University of California San Diego, for sharing resources for adaptive laboratory evolution.

## Author contributions

**Conceptualization:** Amitesh Anand.

**Data curation:** Snehal V. Khairnar.

**Formal analysis:** Snehal V. Khairnar, Amitesh Anand.

**Funding acquisition:** Amitesh Anand.

**Investigation:** Snehal V. Khairnar, Anjali V. Patil, L. Karvannan, Amitesh Anand.

**Methodology:** Snehal V. Khairnar, Amitesh Anand.

**Project administration:** Amitesh Anand.

**Supervision:** Amitesh Anand.

**Visualization:** Snehal V. Khairnar, Anjali V. Patil, Amitesh Anand.

**Writing – original draft:** Snehal V. Khairnar, Amitesh Anand.

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
