## [Decision Letter · Decision Letter 0]

3 Sep 2025

PGENETICS-D-25-00915

Naphthoquinone preferentially pairs with non-proton-pumping NADH dehydrogenase for respiratory electron transport

PLOS Genetics

Dear Dr. Anand,

Thank you for submitting your manuscript to PLOS Genetics. After careful consideration, we feel that it has merit but does not fully meet PLOS Genetics's publication criteria as it currently stands. The reviewers were very positive, but have a few minor comments and suggestions that we ask you to consider. Therefore, we invite you to submit a revised version of the manuscript that addresses the points raised during the review process.

Please submit your revised manuscript within 30 days Oct 03 2025 11:59PM. If you will need more time than this to complete your revisions, please reply to this message or contact the journal office at plosgenetics@plos.org. Please include the following items when submitting your revised manuscript:

We look forward to receiving your revised manuscript.

Kind regards,

Danielle A. Garsin

Section Editor

PLOS Genetics

Danielle Garsin

Section Editor

PLOS Genetics

Aimée Dudley

Editor-in-Chief

PLOS Genetics

Anne Goriely

Editor-in-Chief

PLOS Genetics

**Journal Requirements:**

https://journals.plos.org/plosgenetics/s/submission-guidelines#loc-parts-of-a-submission

5) Please ensure that the funders and grant numbers match between the Financial Disclosure field and the Funding Information tab in your submission form. Note that the funders must be provided in the same order in both places as well.

**Reviewers' comments:**

Reviewer's Responses to Questions

**Comments to the Authors:**

Reviewer #1: The manuscript by Khairnar et al. describes an in-vitro evolution study on the strategies of E.coli strains deleted in both respiratory quinones (low-potential naphthoquinones and high-potential menaquinones) to suppress the challenges generated by the compromised respiratory chain. Naphthoquinone-deletion was found to be leaky due to the presence of an iso-chorismate synthase. Strains were found to recover increased rates of respiration via routing electrons from NADH through the non-proton-pumping NDH-2 enzyme rather than through Complex I. The experiments are exhaustive and well-controlled and the results are presented in a concise but complete manner. The obtained conclusions are extremely pertinent to our understanding of the thermodynamic constraints guiding evolution of energy converting electron transfer chains. I therefore recommend publication of this manuscript in PloS Genetics. I have only two minor comments:

- line 71: “… ndh (gene encoding…” must read “… ndh gene (encoding …”

- lines 120-129. The authors mainly attribute the reason for favoring NDH-2 over Complex I to the higher catalytic efficiency of the former thereby suppressing ROS generated by semiquinone radicals. There are two things to take into account in this context. (a) NDH-2 performs a 2-electron reduction from NADH to quinone via its highly cooperative flavin cofactor while the reduction from the iron-sulfur-center-chain to quinone in Complex I necessarily has to go one-by-one. This implies that basically no semiquinone radical is produced by NDH-2 while Complex I has been shown by a host of experiments to be one of the two major sources together with the Qo-site of Complex III) of ROS in respiratory chains. (b) The driving deltaG from NADH to naphthoquinones is much lower than that to benzoquinones. Since Complex I does additional work (pumping protons and generating a pmf), its function becomes quickly compromised in the presence of the produced pmf. It therefore will stall (the reaction NADH –> quinone + pmf will go to equilibrium) and no electrons will reach the downstream ETS enzymes. This problem is much less pronounced for the benzoquinones due to the increase in deltaG resulting from their higher redox potential.

A short mentioning of these facts in this paragraph might be warranted.

Reviewer #2: This manuscript investigates the metabolic adaptation of naphthoquinone (NQ)–dependent Escherichia coli, with a particular focus on the electron transport chain (ETS). The authors performed adaptive laboratory evolution (ALE) of E. coli strains lacking menF and ubiC, two genes essential for respiratory quinone biosynthesis. They subsequently analyzed mutations in evolved clones and identified upregulation of ndh, encoding the non-proton-pumping NADH dehydrogenase, suggesting its preferential pairing with NQ. Overall, the study is well-executed, and the results are clear and compelling. However, several issues, particularly related to writing style and clarity, need to be addressed before publication. Once these concerns are resolved, I would be happy to recommend the manuscript for publication in PLOS Genetics.

Major Comments

1. Clarity and contextualization of prior work

The text is sometimes overly abstract, particularly when introducing previous studies. Please provide more detailed explanations so that readers can follow without relying heavily on cited literature. For example:

o Line 39: Clarify the specific observation with evolved NQ-dependent E. coli. The sentence “The growth improvement of the NQ-dependent strain imparted by the loss of function of PdhR creates a mechanistic paradox” is vague. Please state what the actual observation was.

o Line 43: Explain the mechanistic relationship between PdhR and NDH-2. Also clarify what “this study” refers to in this context.

o Line 46: Revise to explicitly state the roles of ubiC and menF in NQ and UQ biosynthesis, and confirm that ubiquinones (UQs) are the canonical respiratory quinones.

o Line 59: Clearly indicate that “u” and “e” before strain genotypes designate “unevolved” and “evolved,” respectively.

o Abstract: The sentence “Using adaptive laboratory evolution of a naphthoquinone-dependent Escherichia coli strain, we previously showed the fitness advantage of high-potential quinones” requires elaboration. Please explain what is meant by the energy potential of quinones and why this distinction matters.

o Line ~120: Expand the description of the custom hidden Markov Model profiles: what genes were targeted, what metabolites/enzymes they correspond to, and why these models were developed.

2. Figure 2 and evolved lineages

The authors obtained four independently evolved strains, but only the lineage A isolate was studied in depth. Were additional mutations (beyond the ndh upstream region) identified in the other lineages? Please elaborate on whether parallel or distinct adaptive mechanisms were observed.

3. Overview of ETS and quinone coupling

For clarity, please expand on how quinones interact with the electron transport chain. A concise overview figure or schematic showing the coupling of NQ and UQ with different dehydrogenases and terminal oxidases would strengthen the manuscript.

**Have all data underlying the figures and results presented in the manuscript been provided?**

Reviewer #1: Yes

Reviewer #2: Yes

PLOS authors have the option to publish the peer review history of their article (what does this mean?). If published, this will include your full peer review and any attached files.

Reviewer #1: No

Reviewer #2: No

**Figure resubmission:**
---

## [Editor Report · Decision Letter 1]

10 Sep 2025

Dear Dr Anand,

We are pleased to inform you that your manuscript entitled "Naphthoquinone preferentially pairs with non-proton-pumping NADH dehydrogenase for respiratory electron transport" has been editorially accepted for publication in PLOS Genetics. Congratulations!

Yours sincerely,

Danielle A. Garsin

Section Editor

PLOS Genetics

Danielle Garsin

Section Editor

PLOS Genetics

Aimée Dudley

Editor-in-Chief

PLOS Genetics

Anne Goriely

Editor-in-Chief

PLOS Genetics

Comments from the reviewers (if applicable):

**Data Deposition**

http://datadryad.org/submit?journalID=pgenetics&manu=PGENETICS-D-25-00915R1

**Press Queries**

---

## [Editor Report · Acceptance letter]

PGENETICS-D-25-00915R1

Naphthoquinone preferentially pairs with non-proton-pumping NADH dehydrogenase for respiratory electron transport

Dear Dr Anand,

We are pleased to inform you that your manuscript entitled "Naphthoquinone preferentially pairs with non-proton-pumping NADH dehydrogenase for respiratory electron transport" has been formally accepted for publication in PLOS Genetics! Your manuscript is now with our production department and you will be notified of the publication date in due course.

With kind regards,

Judit Kozma

PLOS Genetics

On behalf of:
